# Anti-Aging Effects of GDF11 on Skin

**DOI:** 10.3390/ijms21072598

**Published:** 2020-04-09

**Authors:** Luc Rochette, Loubna Mazini, Alexandre Meloux, Marianne Zeller, Yves Cottin, Catherine Vergely, Gabriel Malka

**Affiliations:** 1Equipe d’Accueil (EA 7460): Physiopathologie et Epidémiologie Cérébro-Cardiovasculaires (PEC2), Université de Bourgogne—Franche Comté, Faculté des Sciences de Santé, 7 Bd Jeanne d’Arc, 21000 Dijon, France; alexandre.meloux@gmail.com (A.M.); marianne.zeller@u-bourgogne.fr (M.Z.); yves.cottin@ch-dijon.fr (Y.C.); cvergely@u-bourgogne.fr (C.V.); 2Centre interface Applications Médicales CIAM, Université Mohammed VI Polytechnique, Ben-Guerir 43 150, Morocco; loubna.mazini@um6p.ma (L.M.); gabriel.malka@um6p.ma (G.M.); 3CHU Cardiology Unit Dijon, 21000 Dijon, France

**Keywords:** skin aging, regeneration, growth factors, disease

## Abstract

Human skin is composed of three layers: the epidermis, the dermis, and the hypodermis. The epidermis has four major cell layers made up of keratinocytes in varying stages of progressive differentiation. Skin aging is a multi-factorial process that affects every phase of its biology and function. The expression profiles of inflammation-related genes analyzed in resident immune cells demonstrated that these cells have a strong ability to regenerate adult skin stem cells and to produce endogenous substances such as growth differentiation factor 11 (GDF11). GDF11 appears to be the key to progenitor proliferation and/or differentiation. The preservation of youthful phenotypes has been tied to the presence of GDF11 in different human tissues, and, in the skin, this factor inhibits inflammatory responses. The protective role of GDF11 depends on a multi-factorial process implicating various types of skin cells such as keratinocytes, fibroblasts and inflammatory cells. GDF11 should be further studied for the purpose of developing novel therapies for the treatment of skin diseases.

## 1. Introduction

The skin is the largest organ in the human body. Even though it is less than 2 mm thick in most areas, it accounts for about 15% of an adult’s total body weight. In addition to thermoregulation, which is the skin’s primary role, many vital functions are attributed to the skin, including protection from external physical, chemical and biological “aggressors” and prevention of excess water loss. Intrinsic skin aging is an inevitable physiological process; skin cells are constantly shed and then renewed. However, aging impairs skin renewal and is associated with a loss of structural integrity [1]. 

## 2. Skin and Cell Regeneration 

The skin is composed of three layers of tissue: the hypodermis, the dermis and the epidermis. Epidermal cells and dermal fibroblasts play a critical role in defining the skin’s architecture and function. Their mutual interactions are closely related to skin development, homeostasis and repair. Multiple epithelial stem cell (SC) populations also contribute to skin homeostasis. The human epidermis consists of four stratified layers mostly composed of keratinocytes (in various stages of progressive differentiation) and melanocytes. The epidermis is stratified, in ascending order, into basal, spinous, granular, and cornified layers. The dermis makes up most of the skin mass. The structure of the dermis is dense fibroelastic connective tissue that supports extensive vascularity, nerve networks, and specialized sweat glands and hair appendages. The dermis is colonized by fibroblasts surrounded by the components of the dermal extracellular matrix (ECM). Collagen, elastic fibers, glycoproteins, and proteoglycans are present in this matrix. Many genetic and acquired diseases are a result of impaired function of skin ECM or its components [2].

In the skin, integrins are cell surface receptors that mediate cell-to-ECM and cell-to-cell adhesion. These integrins also lead the ECM to physically link the intracellular actin cytoskeleton, thus creating a mechanical force. Integrin αvβ6, which is exclusively expressed in epithelial cells, activates transforming growth factor-β1 (TGF-β1), leading to the modulation of innate immune surveillance of the skin. Interestingly, upregulation of integrin αvβ6 in wounds coincides with regeneration of the basement membrane zone [3]. The basal layer contains mitotically active cells that populate the outer epidermis, which is composed of at least 80% keratinocytes.

The basal layer is considered the headquarters of cell regeneration. This regeneration is accomplished in a hierarchic manner by SCs and transit-amplifying cells. SCs are able to self-renew and are maintained throughout a person’s lifetime. They contribute to epidermal renewal and repair by continuously generating pools of transit-amplifying progenitors [4]. The precise nature of SC division has been studied. The functions of this population of cells have been examined, principally in relationship with the properties of mesenchymal stem cells (MSCs). MSCs are multipotent SCs that have proliferation potential, high self-renewal, and differentiation potential. MSCs are important cells in the skin as they contribute to the ongoing regeneration of the epidermis [5].

The skin is equipped with nerve fibers that convey sensory information for touch, temperature, and pain. These nerves are likely slowly conducting, unmyelinated C-fibers and thinly-myelinated Aδ-fibers. Our sense of touch is controlled by a large system of nerve endings known as the somatosensory system [6]. When the skin is inflamed, keratinocytes secrete mediators that target sensory neurons, immune cells and microvascular endothelial cells. In normal human dermal microvascular endothelial cells, interleukin 8 production increases in response to the neuropeptides released by cutaneous c-fibers [7]. Peripheral neuron regeneration is restricted in patients with damaged or diseased peripheral axons. In cases of cutaneous neurogenic inflammation and local stress (thermal and mechanical), transient receptor potential vanilloid 1 (TRPV1) and ankyrin 1(TRPA1) are known to specifically contribute to pain and are considered to be non-selective cation channels. TRPV1-activation modifies the regenerative process of adult neurons and their axons during epidermal reinnervation [8].

## 3. Skin Aging

Two types of skin aging can be defined: intrinsic (or chronological) aging, and extrinsic aging. Aged skin is characterized by epidermal thinning, wrinkling and a loss of elasticity. The age-dependent remodeling of the dermis is mostly due to the dysfunction of long-lasting resident fibroblast populations. Older fibroblasts lose the ability to structure the ECM, diminishing the production of collagen and elastin. In these conditions, dermal fibroblasts enhance the secretion of angiogenic inducer proteins that promote the secretion of pro-inflammatory cytokines such as interleukin-6 (IL-6) and matrix metalloproteinases (MMPs) such as MMP-9. As the skin ages, pro-apoptotic genes are upregulated as well, thus inducing fragmentation mechanisms that result in functional defects in ECM proteins.

One major extrinsic factor that modifies skin morphology is exposure to UV/infrared (IR) radiation. UV triggers inflammation, immune changes and DNA damage. The altered DNA then promotes cellular senescence and carcinogenesis. Senescent cells increase in number with aging, lose their ability to proliferate, resist apoptosis and secrete factors involved in tissue degeneration [9].

IR radiation can increase reactive oxygen species (ROS) and is involved in different signaling within the skin. Additionally, mitochondria play a major role in the photoaging of human skin, and their activity is reduced in response to IR radiation. Telomeres may be particularly susceptible to oxidative-stress-induced damage, which is slow to repair [10].

In certain cases, the skin might also be physiologically predisposed to accelerated aging and carcinogenesis; this is the case in various genetic syndromes that favor DNA damage or telomere dysfunction and cellular senescence. A decline in the DNA’s ability to repair itself, increasing oxidative stress, shortening of the telomeres, and the production of progerin, may drive cells towards senescence. Progerin, which is a mutant form of the lamin A protein, may be one of several physiological biomarkers of the aging process [11].

Concerning the cellular biology of the skin, evidence indicates that epigenetic processes can reversibly impact skin aging, either through DNA methylation, histone modifications or microRNAs (miRNAs) [12]. Epigenetic code and chromatin status are interconnected and exhibit their effects on cell proliferation and differentiation by regulating the gene expression profile of every single cell. 

## 4. Cutaneous Wound Healing

Following skin injury, stem cells must react quickly to repair tissue and restore the damaged barrier. Cutaneous wound healing requires the complex interplay of four stages, each incorporating different cellular events: immediate injury responses characterized by blood clot formation, inflammatory cell recruitment, re-epithelialization/revascularization and scar remodeling [13]. The inflammatory response to tissue injury is a key process of the wound healing response. Neutrophils circulating in the blood move into the tissue via endothelial attachment and extravasation mechanisms. Multiple growth factors released at the site of tissue injury, such as vascular endothelial growth factor-A (VEGF-A) and platelet-derived growth factor, induce the formation of new blood vessels from remaining endothelial cells. The formation of new blood vessels, also called neovascularization, is an essential process for effective wound healing. It provides optimal distribution of substrates and preservation of oxygen homeostasis, which are good conditions for tissue regeneration [14]. 

When the skin tissue is damaged, mitogenic and other growth-promoting factors are released by activated platelets and ECM storage sites. In the first phase of inflammation, these factors create a proliferative response. Changes also occur in the activation state of certain cells (such as resident macrophages and colonizing monocytes) during inflammatory phenomena and tissue repair. These changes promote angiogenesis, improved epithelial continuity, and growth and differentiation of SCs that are associated with the stimulation of fibroblast activity. Different populations of SCs have various roles in the skin, including controlling inflammation or the healing process, accelerating the migration and proliferation of skin cells, improving angiogenesis and even limiting the signs of aging. In this area, the role of MSCs is important; they are derived from the mesoderm and can differentiate into a variety of tissues [15]. 

The process of tissue regeneration effectively repairs the skin through re-epidermalization, epidermal and stromal cell interactions, and angiogenesis. A variety of cell types, including several SC populations, reinforce the epidermis. One essential characteristic of SCs is plasticity, which denotes the possibility of differentiating into several tissue types, and another key characteristic is self-renewal. Epidermal SCs have essential properties specifically related to proliferation and differentiation that make them a particularly important cell population for skin tissue regeneration. Epidermal SCs are skin stem cells whose origins may be heterogeneous or autogenous. Many studies have explored wound healing therapies that use SCs [16].

Various signaling and transcriptional pathways regulate in a stage-specific manner the expression of genes implicated in epidermal SC properties. Epidermal SCs have been conventionally classified as slow-developing and long-lived cells that are found in certain spots on the skin. Regarding the maintenance and differentiation of epidermal SCs, it has been shown that different signaling pathways appear to be involved, including the Notch, Wnt/β-catenin, and p63 pathways. The Wnt/β-catenin and p63 pathways are central to epidermal lineage selection [17].

Although the essential role of p63 in epidermal biology has been established, the regulatory mechanisms implicated in the properties of p63 are not yet fully understood. The TP63 gene encodes several isoforms of p63 due to the presence of alternative promoters. In human epidermis, ∆Np63 is the predominant isoform and interacts with several transcription factors such as AP-1 and PPAR-alpha.

The Wnt/β-catenin signaling pathway is regulated by different agonists and antagonists [18]. It has been shown that epidermal SCs express specific components of the canonical Wnt pathway. Studies have demonstrated that the regulation of the lineage selection program of epidermal SCs passes through the epidermis Wnt/β-catenin signaling pathway. Many types of epidermal tumors involve Wnt/β-catenin signaling, which suggests that this pathway has a critical role in cutaneous neoplasms.

## 5. Growth Factors and Regenerative Process

It was recently demonstrated that diverse types of blood proteins contribute to the regenerating process via different mechanisms. Together with cytokines, blood proteins are important regulators of cell division, differentiation, cell activation, apoptosis, and cell survival [19]. Proteins with regenerative capacities, such as growth factors (GFs), are central morphogenetic proteins that initiate cell behavior and control tissue repair and renewal. GF proteins are naturally secreted from cells, and they interact with cell surface receptors of the ECM. GFs provide an essential contribution to the regenerative process, and studies on skin regeneration have applied exogenous GFs in an effort to regenerate damaged tissue. 

### 5.1. Epidermal Growth Factor Receptor (EGFR) and Its Ligands

The growth and repair processes of epithelia are recognized to be centrally directed through EGFR and its ligands. Additionally, it has been established that EGFR is highly involved in the control of inflammatory responses. Several skin disorders including psoriasis are related to the anomalous activation of EGFR signaling [20]. The EGFR/ligand system activates various signal transduction pathways, and is influenced by several extracellular stimuli [21]. In the epidermis, EGFR is mainly expressed in proliferating basal keratinocytes and to a lesser degree in suprabasal cell layers. Keratinocyte proliferation and migration is supported by EGFR, which also delays apoptosis in suprabasal keratinocytes that have lost their interaction with the matrix. 

The maintenance of skin tissue size and scar-free skin regeneration are supported by self-organization of skin-dividing SCs. EGF and EGFR play an essential role in stimulating the dermal and epidermal regeneration necessary for wound healing. A large number of EGFR ligands are produced by epidermal keratinocytes, including TGF-α, heparin binding (HB)-EGF, amphiregulin, and epiregulin. In chronic inflammatory skin disorders such as psoriasis [22], there is an upregulation of EGFR and expression of its ligands, whereas in fibrotic skin scarring, enhanced TGF-β expression has been observed. Consequently, the inhibition of TGF-β is believed to exert “antiscarring” properties, and members of the TGF-β superfamily have a significant role in scar-free skin regeneration. The most important regulator of skin wound healing is certainly TGF-β1 in association with connective tissue growth factor (CCN2/CTGF), although both TGF-β1 and TGF-β3 might be involved in wound healing [23]. 

### 5.2. Brain-Derived Neurotrophic Factor (BDNF) and Insulin-Like Growth Factors (IGFs) Trophic Factors with “Yin and Yang” Effects on the Skin (Figure 1)

The group of neurotrophins includes various growth factors such as nerve growth factor (NGF) and BDNF. Each of these molecules has a prodomain that is cleaved to yield the mature isoform. Many stimuli, such as hormones, exert temporal control over BDNF transcription. Two receptors have been identified for BDNF: tropomyosin receptor kinase B (trkB) and the common neurotrophin receptor, p75NTR. The mature form of BDNF preferentially binds to trkB, resulting in pro-growth signaling. However, proBDNF preferentially binds p75NTR, resulting in antigrowth signaling. The two receptors for BDNF have opposing roles and maintain a balance between growth and death. BDNF binds to a p75NTR-sortilin complex. As a neurotrophin, BDNF has emerged as an important regulator of axon regeneration in skin. p75NTR, the receptor for BDNF, is expressed in sensory neurons. After skin injury, sensory neurons decreased expression of p75NTR, which could act as a survival signal [24]. Recent results show some relationship between BDNF and other factors such as growth differentiation factor 11 (GDF11) and IGFs. GDF11 enhances neurogenesis and angiogenesis by regulating the GDF11 and TGF-β/Smad2/3 signaling pathways [25]. Other types of growth factors also play a central role in regulating cell proliferation, differentiation and apoptosis in various tissues. For instance, IGFs interact with specific glycoprotein membrane receptors: type I (IGF-1R), type II (IGF-2R), insulin receptor (IR) and hybrid receptors (IGF-1R/IR). The importance of the IGF system, in particular IGF-I, was demonstrated for the acute photo-response in keratinocytes [26].

Since its discovery, NGF has occupied a critical role in developmental and adult neurobiology because of its many important regulatory functions relative to the survival, growth and differentiation of nerve cells. Studies in humans revealed that topical administration of NGF was a promising approach for the treatment of cutaneous pressure ulcers [27], and the topical application of NGF may also represent a new useful tool for the management of difficult diabetic ulcers or severe pressure ulcers [28,29].

It appears that disturbances in IGF signaling pathways are involved in several skin disorders, in particular epidermal hyperplasia. IGF-1 plays a substantial role in keratinocyte survival and exerts power over melanogenesis, which is affected in vitiligo ^30^. IGF-1 deficiency results in vascular cells that are less able to maintain an effective Nrf2-dependent antioxidant defense system in response to increased oxidative stress. IGF-1 is at the crossroads of several GH responses and is able to activate multiple signaling cascades, resulting in a potent proliferative signal [30].

## 6. Potential Activity of Endogenous Factors on Skin Regeneration: Role of GDF11 

### 6.1. Structure and Formation of GDF11

GDF11 regulates essential cell differentiation and proliferation responses [31,32]. GDF11, also known as bone morphogenic protein 11 (BMP-11), is a member of the BMP/transforming growth factor β (TGF-β) family, and it plays a key role in the growth and development of several species, including humans. GDF11 is produced from a precursor protein by proteolytic processing and is expressed in numerous tissues, including the skin, heart, skeletal muscle, and developing nervous system. Its expression is at the highest level in young adult organs and seems to decline during aging [33]. TGF-β family ligands such as GDF11 bind and activate specific heteromeric type I and type II Ser/Thr kinase receptor complexes, which transmit signals by phosphorylating receptor regulated (R)-Smads. Two distinct R-Smad pathways exist: the TG-Fβ-Smad pathway (R-Smad2/3) and the BMP-Smad pathway (R-Smad1/5/8). Members of the Smad family of transcription factors are important intracellular messengers. The type I receptor subsequently phosphorylates Smad 2 or 3, allowing this complex to associate with Smad 4 and then to transfer into the nucleus. This Smad complex, Smad2/3/4, is then localized to the nucleus.

At first, GDF11 binds to activin receptor II (ActRII), including ActRIIA and ActRIIB, and then recruits activin receptor I (ActRI) (including ALK4, and ALK5) (Figure 2). Transcription factor members of the Smad family are important intracellular messengers. TGF-β signaling is strongly controlled by post-translational modifications, which regulate the stability, subcellular localization, and activity of the signaling components [34]. The GDF11 gene was mapped to human chromosome 12q13.2. It encodes a 407 amino acid protein with a secretion signal sequence. The Gdf11 promoter region can be activated by trichostatin A, an antibiotic which inhibits histone deacetylase (HDAC) enzymes [35]. Some studies have shown that GDF11 can reverse age-related dysfunction in muscle, nervous and cardiovascular systems. Although serum GDF11 levels were found to be decreased in old mice, supplementation with GDF11 “rejuvenated” them, thereby suggesting that GDF11 is a key player in mammalian life span. It has also been suggested that GDF11 is involved in the age-related global physiological decline in function, and that restoring circulating blood levels of GDF11 could reverse some of the cellular and physiological dysfunctions observed in aged mice [33].

### 6.2. Effects of GDF11 on Skin Regeneration

The search for humoral factors that are able to affect regenerative capacity of the skin, such as GDF11, should be re-examined. In tissues, GDF11 has demonstrated its ability to regulate progenitor differentiation and/or proliferation. 

It is well known that keratinocytes are modified as they move between the epidermal layers towards the surface of the body. Spindle alignment is controlled by intrinsic and extrinsic factors including cell polarity. During the differentiation of keratinocytes, the precise regulation of the process is strictly regulated by different signaling pathways, which are controlled by Ca^2+^, cell–cell contact, and TGFβ. The role of GDF11 appears through the expression of factors such as p27^Kip1^ and IL-23 (Figure 1).

The effect of GDF11 on progenitor cells is associated with its ability to upregulate p27^Kip1^ [36]. P27^Kip1^ is a key regulator of cell growth arrest and terminal differentiation in keratinocytes [37]. In a human psoriatic keratinocyte cell line, it has been shown that the P27^Kip1^ protein expression level was decreased in both cytoplasmic and nuclear compartments, suggesting a translocation from the nucleus and subsequent degradation of this protein. P27^Kip1^ significantly contributes to psoriasis pathogenesis characterized by dysregulated keratinocyte proliferation. Psoriatic epidermal hyperplasia induced by aberrant keratinocyte differentiation and resistance to apoptosis may be due to the downregulation of p27^Kip1^, which is able to promote apoptosis [38]. The relationship between GDF11 and IL-23 expression in keratinocytes is suggested by skin inflammation. An IL-23-induced skin inflammation model that was established in order to investigate the anti-inflammatory effect of GDF11 found that, on skin, the infiltration of inflammatory cells and epidermal thickening were limited by recombinant GDF11 (rGDF11). According to recent studies, GDF11 may represent a promising target for the prevention and treatment of psoriasis-like skin [39].

As previously reported, fibroblast implantation plays a crucial role in skin regeneration and healing of wounds; human dermal skin fibroblasts are important for wound healing because they secrete type I collagen and cytokines. In this context, the role of GDF11 in skin dermal fibroblasts has been investigated. GDF11 expression and activity were decreased in fibroblasts originating from adult donors as compared to neonatal fibroblasts. The induction of collagen I and III was the main effect of GDF11, in both adult and neonatal fibroblasts, by triggering Smad signaling in a TGF-β-like fashion [40].

Concerning the connection between MSCs and skin regeneration, recent studies suggest that MSCs secrete cytokines and growth factors that are implicated in regeneration and wound healing. Primitive umbilical-cord-blood-derived MSCs are responsible for GDF11 secretion. In fibroblasts, GDF11 stimulates the secretion of ECM proteins including collagen type I and III and fibronectin [41]. 

Tumor necrosis factor-α (TNF-α) plays a key role in inflammatory diseases, including skin inflammation, while GDF11 inhibits inflammatory reactions. GDF11 treatment antagonizes TNF-α-induced inflammation in macrophages [42], and the administration of GDF11 appears to attenuate skin inflammation. Studies show that TNF-α–induced activation of the nuclear factor kappa B (NF-κB) signaling pathway, which is known to participate in various inflammatory conditions, is limited by GDF11 treatment [39]. It is known that macrophages are closely associated with inflammatory reactions including psoriasis-like skin inflammation. Psoriasis is the typical reaction of skin that is infiltrated by specific immune cells implicated in inflammation and which result in the destruction of the outer layer of the skin. In models of psoriasiform in mice, rGDF11 treatment reduced the accumulation of macrophages in the skin tissue by signifying the reduction of inflammatory cell infiltration. In vivo, these effects were associated with the inhibition of the expression of inflammatory cytokines such as IL-1β, and IL-6. As we previously reported, GDF11 recruits ActRI including ALK4 and ALK5. The role of activins in the process of skin repair was demonstrated through the regulation of skin properties and immune cell migration [43]. Another recent study [44] looked at the effect of rGDF11 in various skin cells such as human epidermal dermal fibroblasts, keratinocytes, melanocytes, dermal microvascular endothelial cells and 3D skin equivalents, as well as in ex vivo human skin explants. When the skin models were treated with physiologically relevant levels of rGDF11, researchers saw substantial changes in the production of hyaluronic acid and procollagen I. This study established that rGDF11 was able to induce Smad2/3 phosphorylation in those cells, inducing possible beneficial effects on skin vasculature, which is altered by aging [45].

## 7. Conclusions and Future Directions 

Finally, injured skin is able to spontaneously self-repair, a process which is mediated by numerous pleiotrophic growth factors including members of the TGFβ and VEGF families. Before, during and after injury, epidermis keratinocytes express a large panel of growth factor ligands and receptors, including VEGF, VEGFR1, VEGFR2, phosphorylated Smad2, and TGFβ1, and activins [46]. As a member of the TGF-β superfamily, GDF11 activates the TGF-β signaling pathway by phosphorylating Smad2/3. It is widely known that the Smad2/3 and Akt serine/threonine kinases are implicated in signal transduction and gene expression. The phosphatidylinositol 3-kinase (PI3K)/AKT pathway is involved in multiple biological processes in the skin in connection with the production of heat shock proteins (HSPs). HSP27, HSP70 and HSP90 show different patterns of expression in human keratinocytes. HSPs are molecular chaperones essential for the maintenance of cellular functions, but they can be released extracellularly upon cellular injury or necrosis [47]. GDF11 induces protective effects in various tissues through the suppression of oxidative stress and the expression of HSPs; the AKT/Smad 2/3 pathways are also implicated in these events (Figure 1 and Figure 2). As the key member of the TGF-β superfamily, GDF11 represents a promising therapeutic agent for the treatment of a number of inflammatory skin diseases, including psoriasis.

## Figures and Tables

**Figure 1 ijms-21-02598-f001:**
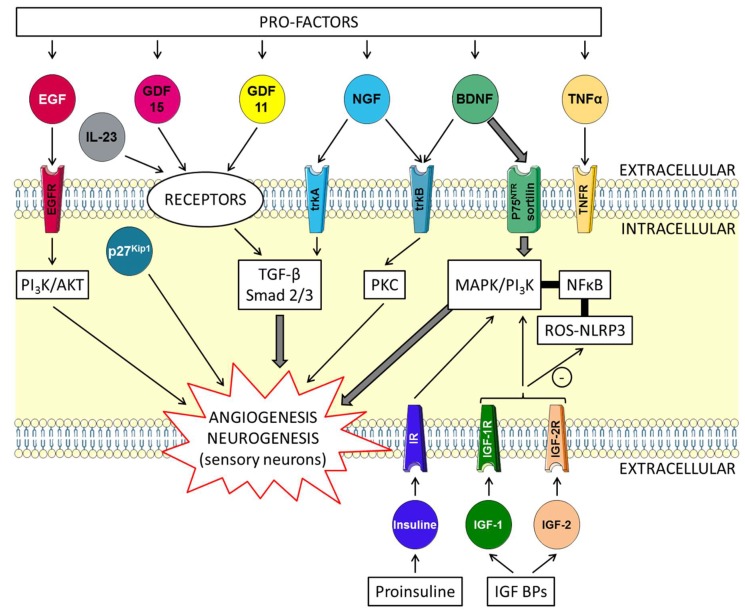
Neurotrophins and their effect on angiogenesis and neurogenesis in the skin. Brain-derived neurotrophic factor (BDNF) binds to two receptors—tropomyosin receptor kinase B (trkB) or the neurotrophin receptor, p75^NTR^. BDNF preferentially binds to a P75 ^NTR^-sortilin complex. TrkB can activate various intracellular pathways, such as the protein kinase C (PKC). Nerve growth factor (NGF), growth differentiation factor-11 (GDF11) and growth differentiation factor-15 (GDF15) act on neurogenesis and angiogenesis through the TGF-β/Smad2/3 signaling pathway. Insulin and insulin-like GFs (IGFs) bind to membrane receptors: type I (IGF-1R), type II (IGF-2R), insulin receptor (IR) targeting MAPK and PI3K. Bioavailability of the IGFs is regulated by specific binding proteins (IGFBPs). IGFs affect multiple signaling cascades through reactive oxygen species (ROS) metabolism and the critical regulator of inflammation NLRP3. P27^Kip1^ is a key regulator of cell growth arrest and IL-23 expression in keratinocytes is associated with inflammation. Epidermal growth factor receptor (EGFR) and its ligands (EGFR) stimulate the AKT/PI3K pathway. Tumor necrosis factor-α (TNF-α) induces activation of the nuclear factor-kappa B (NF-κB) signaling pathway limited by GDF11.

**Figure 2 ijms-21-02598-f002:**
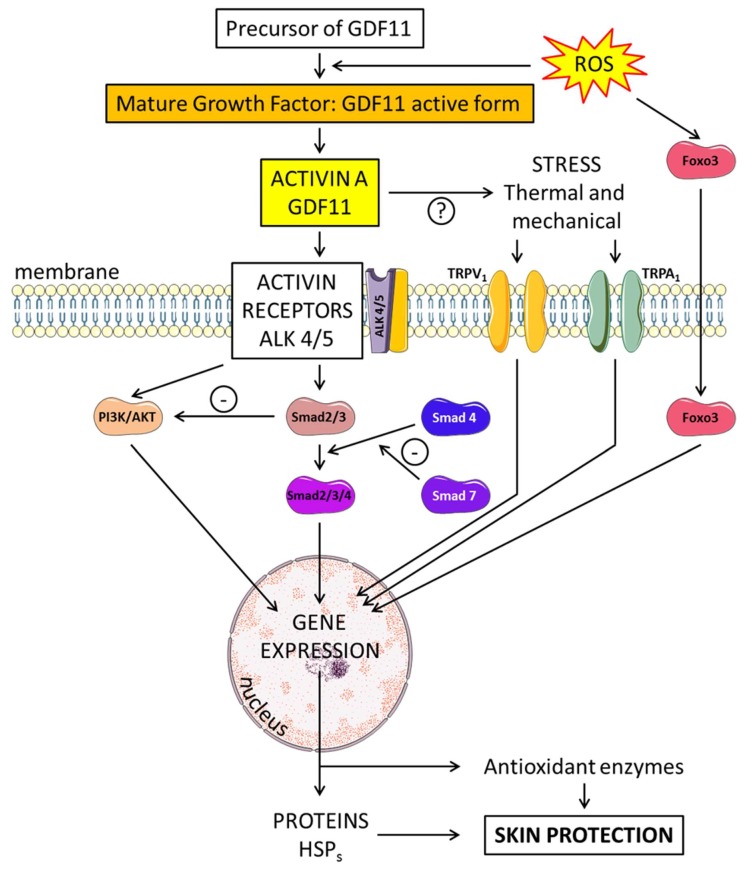
Schematic overview of the skin-protecting properties of GDF11. Mature GDF11 binds to activin receptor II (ActRII), including ActRIIA and ActRIIB, and then recruits activin receptor I (ActRI), including ALK4, and ALK5. Members of the Smad family of transcription factors are important intracellular messengers. The type I receptor subsequently phosphorylates Smad 2 or 3, allowing this complex to associate with Smad 4. This Smad complex, Smad2/3/4, is then localized to the nucleus. The Smad complex acts in the nucleus on target genes regulating transcription. The phosphatidylinositol 3-kinase (PI3K)/AKT pathway (PI3K/AKT), p27kip1and IL-23 are implicated. Proteins (heat shock proteins: HSPs) and antioxidant enzymes are produced. Foxos translocate to the nucleus where they activate programs of gene expression. Foxo3 is involved in counteracting oxidative stress. Gene expression is modified after stimulation of the transient receptor potential vanilloid 1 (TRPV1) and ankyrin 1 (TRPA1) channels.

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
