# Peer review of "Anti-Aging Effects of GDF11 on Skin"

_ijms, 2020, doi:10.3390/ijms21072598_

Round 1
Reviewer 1 Report
This is a review article that summarizes the role of GDF11 in skin diseases. The article is well-written and thorough. I think it should be clarified that this is a review article, which is not apparent as it is, so that the readers do not expect any new results. I think the work is of use to the community, but there is nothing particular novel in the work. I have only a few minor comments that should be addressed:
Lines 140-148: Add a reference to this part of the paragraph.
Figure 1: Make sure that all labels/abbreviations are explained and that they are written the same way in the figure as in the caption.
Figure 2: "The type I receptor subsequently phosphorylates Smad 2 or 3, allowing this complex to associate with Smad 4 and then to transfer into the nucleus." Is Smad 2/3/4 a complex between Smad 2 or 3 with Smad 4? This needs to be clarified.
Abbreviations: ACT - activin
Author Response
The authors thank the Referee for the careful review of our manuscript and for the interesting suggestions he/she made. We have modified and/or removed some parts.
I have only a few minor comments that should be addressed:
Lines 140-148: Add a reference to this part of the paragraph
We added a recent reference:
Mariana Barreto Serra and al. From Inflammation to Current and Alternative Therapies Involved in Wound Healing. 2017 Int J Inflam, 340621
Figure 1: Make sure that all labels/abbreviations are explained and that they are written the same way in the figure as in the caption.
We added two points:
P27Kip1 is a key regulator of cell growth arrest
IL-23 expression in keratinocytes is associated with inflammation.
Figure 2: "The type I receptor subsequently phosphorylates Smad 2 or 3, allowing this complex to associate with Smad 4 and then to transfer into the nucleus." Is Smad 2/3/4 a complex between Smad 2 or 3 with Smad 4? This needs to be clarified.
The manuscript has been modified in order to clarify the properties of Smad
6.1: “Members of the Smad family of transcription factors are important intracellular messengers. The type I receptor subsequently phosphorylates Smad 2 or 3, allowing this complex to associate with Smad 4 and then to transfer into the nucleus. This Smad complex: Smad2/3/4 is then localized to the nucleus”
Fig2: The type I receptor subsequently phosphorylates Smad 2 or 3, allowing this complex to associate with Smad 4 . This Smad complex: Smad2/3/4 is then localized to the nucleus.
Reviewer 2 Report
While the submitted manuscript is well organized and written, there are a few points of revision before accepted.
First of all, while the title of manuscript is about the "protective roles of GDF11 in skin" there are too small number of studies supporting the authors' rationale. Actually, there are only one reference suggesting direct involvement of GDF11 on skin inflammation. As a translational research, it may be reasonable to postulate that DGF11 plays anti-inflammatory roles in skin but without sufficient evidences, it should be more cautious for choosing a manuscript title.
In the meantime, cause there a few studies suggesting the potential anti-aging effects of GDF11 on skin aging, while still not sufficient and without clinical efficacy, it would be better to change the title implicating the anti-aging effects of GDF11, rather than protective roles.
Also, the first part of the manuscript covering skin structure, aging, and wound healing is too long, compared with the main part discussing growth factors and skin. It would be better to revise the manuscript into more concise one and focusing on the roles of growth factors and more specifically GDF11 in skin homeostasis.
Author Response
The authors thank the Referee for the careful review of our manuscript and for the interesting suggestions he/she made. We have modified some parts
This new review has been redacted in order to make the findings in specific fields more clear.
…, it would be better to change the title implicating the anti-aging effects of GDF11, rather than protective roles.
The tile has been modified in order to clarify the properties of GDF11
NEW TITLE: Anti-aging effects of GDF11 on skin
The first part of the manuscript covering skin structure, aging, and wound healing is too long, compared with the main part discussing growth factors and skin. It would be better to revise the manuscript into more concise one and focusing on the roles of growth factors and more specifically GDF11 in skin homeostasis.
In agreement with yours comments, the first part of the manuscript has been modified in order to integrate the properties of GDF11 in skin homeostasis.
Some parts of the first manuscript were deleted.
For ex:
2.1. Spinous Layer
The thickness of the spinous layers depends on body site location. Desmosomes ensure intercellular connections between spinous cells and between epidermis cells, promoting mechanical resistance to physical stress. The cells of the epidermis are also connected by gap junctions, such as intercellular channels formed by connexin proteins, that allow the electrical and biochemical signals to travel between cells. Mutations in the encoding of connexins are directly responsible for several autosomal dominant genetic epidermal diseases 4. Chemical signals also help with the regulation of cell metabolism and differentiation through the physiological communication that is ensured by these junctions.
Round 2
Reviewer 2 Report
The manuscript is revised appropriately and reviewer has no further comments.